# A Holistic Strategy of Mother and Child Health Care to Improve the Coverage of Routine and Polio Immunization in Pakistan: Results from a Demonstration Project

**DOI:** 10.3390/vaccines12010089

**Published:** 2024-01-16

**Authors:** Muhammad Atif Habib, Sajid Bashir Soofi, Zamir Hussain, Imran Ahmed, Rehman Tahir, Saeed Anwar, Ahmed Ali Nauman, Muhammad Sharif, Muhammad Islam, Simon Cousens, Zulfiqar A. Bhutta

**Affiliations:** 1Centre of Excellence in Women and Child Health, Aga Khan University, Karachi 74800, Pakistan; habib.atif@yahoo.com (M.A.H.); sajid.soofi@aku.edu (S.B.S.); imran.ahmed@aku.edu (I.A.); 2Department of Pediatrics & Child Health, Aga Khan University, Karachi 74800, Pakistan; 3Trust for Vaccines and Immunization, Karachi 74400, Pakistan; zamir.suhag@tvi.org.pk (Z.H.); rehman.tahir@tvi.org.pk (R.T.); 4Prime Institute of Public Health, Peshawar 25160, Pakistan; sanwar@piph.prime.edu.pk (S.A.); anauman@piph.prime.edu.pk (A.A.N.); msharif@piph.prime.edu.pk (M.S.); 5Centre for Global Child Health, The Hospital for Sick Children, Toronto, ON M5G 1X8, Canada; muhammad.islam@sickkids.ca; 6London School of Hygiene and Tropical Medicine, London WC1E 7HT, UK; simon.cousens@lshtm.ac.uk

**Keywords:** childhood immunization, polio eradication, mother and child health, community engagement, Pakistan

## Abstract

Background: The eradication of poliovirus and improving routine immunization (RI) coverage rates present significant challenges in Pakistan. There is a need for interventions that focus on strengthening community engagement to improve routine immunization coverage. Our primary objective is to assess the impact of an integrated strategy designed to enhance community engagement and maternal and child health immunization campaigns on immunization coverage in Pakistan’s high-risk union councils of polio-endemic districts. Method: We implemented an integrated approach for routine immunization and maternal and child health in the polio-endemic district of Pakistan. This approach involved setting up health camps and actively engaging and mobilizing the local community. An independent team conducted surveys at three key points: baseline, midline, and endline, to evaluate immunization coverage among children under the age of five. The primary outcome measures for the study were coverage of OPV, IPV, and changes in the proportion of unvaccinated and fully vaccinated children. To select clusters and eligible households in each cluster, we utilized a 30 × 15 cluster sampling technique. Multivariable associations between socio-demographic factors and changes in the proportion of fully vaccinated children at the UC level were assessed using hierarchical linear regression models. Results: A total of 256,946 children under the age of five (122,950 at baseline and 133,996 at endline) were enrolled in the study. By the endline, full immunization coverage had increased to 60% or more in all three study areas compared to the baseline. Additionally, there was a significant increase in the coverage of both OPV and IPV across all three provinces at the endline. The full immunization rates were assessed on three levels of the framework: the distal, intermediate (access and environment), and proximal level (camp attendance and effectiveness). At the distal level, on multivariate analysis, family size was found to be a significant predictor of change in immunity within the families (β = 0.68; *p* ≤ 0.0001). At the intermediate level, the likelihood of full immunization decreased with the decrease in knowledge about vaccination (β = −0.38; *p* = 0.002), knowledge about polio vaccine (β = −0.25; *p* = 0.011), and knowledge about IPV (β = −0.06; *p* = 0.546). Perceived obstacles to vaccination were fear of adverse events (β = −0.4; *p* ≤ 0.0001) and lack of education (β = 0.23; *p* = 0.031), which were found to be significant in bivariate and multivariate analyses. At the proximal level, community mobilization (β = 0.26; *p* = 0.008) and attendance at health camp (β = 0.21; *p* ≤ 0.0001) were found to enhance full immunization coverage. On the other hand, the most prominent reason for not attending health camp included no need to attend the health camp as the child was not ill (β = −0.13; *p* = 0.008). Conclusions: This study found that community mobilization and attendance at health camps significantly enhanced full immunization coverage. The findings highlight the importance of community engagement and targeted interventions in improving immunization coverage and addressing barriers to healthcare seeking.

## 1. Introduction

Poliomyelitis has been eradicated almost everywhere except a few hotspots in Afghanistan and Pakistan [1]. Globally, cases of poliomyelitis have decreased by over 99%, since 1988 from approximately 350,000 reported cases to a mere 20 cases in 2022 [2]. In Pakistan, one of the two residual reservoir countries with wild polio, the caseload has fluctuated greatly with 12 polio cases reported in 2018, 147 in 2019, 84 in 2020, 1 in 2021, 20 in 2022 and 3 cases to date in 2023 [3]. Despite the progress in reducing polio cases and their geographical spread, poor access to immunization, widespread malnutrition, and insecurity contribute to remaining pockets of circulating poliovirus, which remain a significant challenge for global polio eradication [1,4]. Failure to eradicate polio could result in a global resurgence; hence, a final push towards eradication must be made [2].

In Pakistan, routine immunization (RI) against poliovirus and other diseases was initiated in 1978 with the National Expanded Program of Immunization (EPI) launch. RI is mostly delivered at fixed immunization centers with minimal outreach services that the vaccinators provide. The coverage rates for RI have remained low, between 65% and 73%, as indicated by the literature [5,6]. Implementing Supplementary Immunization Activities (SIAs) in addition to four doses of routine OPV within RI services has been the principal strategy to enhance OPV coverage. These SIAs are organized by the Pakistan National Polio program at the household level and usually last for 5 days, with 3 days dedicated to visiting all households with children under 5 years of age and the remaining 2 days used to revisit households where children were initially absent or unavailable [7]. Such door-to-door visits to raise population immunity were sufficient to halt poliovirus transmission. Under the national EPI schedule in Pakistan, children receive Bacille Calmette–Guérin (BCG), the first dose of oral polio vaccine (OPV0), and HepB shortly after birth. At 6 weeks, the schedule includes OPV1, Rotavirus1, first dose of pneumococcal conjugate vaccine (PCV1), and pentavalent. At 10 weeks OPV2, the schedule includes Rotavirus2, PCV2, and Pentavalent2. At 14 weeks, the children are given OPV3, the first dose of inactivated polio vaccine (IPV1), PCV3, and Pentavelannt3. At 9 months of age, the child receives the first dose of measles-containing vaccine (MCV1), IPV2, and typhoid. Finally, at 15 months, MCV2 is being given [8].

Pakistan faces several challenges that hamper the success of polio-eradication initiatives. These include issues of vaccine availability, accessibility, acceptance and quality of vaccines, misconceptions about OPV, the killing of polio workers, military-action-related curfews, geographical limitations, restrictions in reaching the population, migration, and refusals [9,10,11]. These barriers are compounded by inadequacies within the RI program, high burdens of childhood diarrhea, malnutrition, potentially impairing intestinal mucosal and systemic response to OPV, as well as poor environmental sanitation, which increases the risk of circulating wild polioviruses and the risk of polio transmission [12]. The country also faces the challenge of introducing inactivated polio vaccine (IPV) along with the third dose of pentavalent vaccine and the coordinated replacement of trivalent OPV by bivalent OPV [13,14]. IPV introduction addressed the risk of the emergence of circulating vaccine-derived polioviruses (cVDPV), typically cVDPV2, which provides an enhanced serum antibody response and could potentially improve mucosal immunity, along with a substantial reduction in excretion of poliovirus when a combination of OPV and IPV is given instead of OPV alone [14,15,16,17]. However, in the presence of low RI coverage, limited access, and minimal community engagement, initiation of IPV also posed numerous challenges in conflict-affected locations of the country where OPV refusals were already common and further worsened by external factors, including political or intelligence campaigns, which have impacted public trust in immunization programs [1,10].

In the South Asian region, an estimated 3 million children under one year of age remain unvaccinated according to the 2022 estimates from the WHO and UNICEF [18]. Bangladesh stands out with the highest proportion of children receiving complete childhood immunization (88.2%), followed by Nepal (79.2%), India (77.1%), and Pakistan (76.5%) [19]. On the other hand, Afghanistan reports the lowest immunization rates at 42.6% [19]. Notably, when considering POV3, all South Asian countries reported coverage exceeding 90%, except for Afghanistan, Nepal, and Pakistan [18].

Notwithstanding these challenges, Pakistan did make important progress in its journey towards stopping poliovirus transmission over the past two decades. This progress was the result of concerted efforts by the government, civil society, and all national and international partners who coordinated and implemented a national polio emergency action plan, aimed at overcoming long-standing challenges during polio immunization activities, particularly in the high-risk reservoir areas of Peshawar, Karachi, and Quetta [20]. However, building trust and demand for polio vaccines requires great attention to effective communication and community mobilization. This includes reframing polio messages within the broader context of preventive health services for children (polio vaccines, RI, exclusive breastfeeding, hand washing, and diarrhea management) [21].

This challenge of implementing a community mobilization strategy and community-based maternal and child health camps, including administration of RI and IPV, was tested through a large cluster randomized controlled trial (cRCT) in several insecure areas of Karachi, the Sindh–Baluchistan border region, and a district in Khyber Pakhtunkhwa [22]. The trial proved that despite misgivings, it was possible to engage the community, and achieve a high level of acceptability for OPV and IPV in the community with 80% coverage at all study sites and an increase in RI coverage with no adverse events. The community responded to the participatory process, and the holistic approach to MNCH with OPV/IPV included within the RI framework. Although there were challenges in accessing certain areas, these were overcome by involving local religious leaders and including local facilitators and volunteers in the team. The cRCT showed a significant drop in refusals and the health camps were a popular site for delivering integrated MCH interventions, especially EPI vaccines. However, concern remained with regards to the scalability of this approach to other insecure settings.

We undertook an extended project in 146 high-risk union councils in endemic polio-endemic districts of Sindh, FATA agencies and Khyber Pakhtunkhwa between 2014 and 2016, in close coordination with the local government, WHO and UNICEF Staff with the explicit aim of delivering an integrated strategy for scaling up MNCH and RI services (including OPV and IPV) using community mobilization and camps.

This study evaluates the impact of an integrated strategy designed to enhance community engagement and maternal and child health immunization campaigns in the high-risk union councils of polio-endemic districts in Pakistan.

## 2. Methods

### 2.1. Study Design and Setting

We used an approach based on community engagement and a human-centric design that considered common misperceptions and barriers related to maternal and child healthcare seeking, specifically childhood immunizations. As with the cRCT [22], polio immunization was included in the context of RI, focusing on complete immunizations. Figure 1 details the approach taken in the design process for the intervention and community mobilization strategy.

We implemented the integrated intervention in 146 union councils of 10 high-risk districts of Karachi in Sindh, Khyber Pakhtunkhwa (KP), and Quetta block districts in Baluchistan between 2014 and 2016 (Figure 2).

### 2.2. Sampling Technique

We evaluated the impact of the intervention through independent baseline, endline, and post vaccination surveys on a representative population. To assess the impact and coverage of the intervention, we planned to collect the data through cross-sectional surveys using a 30 × 15 cluster sampling technique, a standard approach for collecting immunization coverage data. We selected 30 vaccination areas based on proportion to population size PPS. We conducted a fresh line listing of these 30 areas and selected 15 HH with an under-five child from each area randomly through a computer program. A total of 3 cross-sectional surveys were conducted: one before the health camps began, one in between (midline), and one at the end as an endline survey. The study was conducted from 2014 until 2016.

### 2.3. Procedure

We undertook a baseline survey to collect demographic, socioeconomic, routine immunizations, and health-seeking data in the study areas. The baseline survey was conducted by locally recruited and trained data collection teams, who were strictly independent of the implementing teams. A structured questionnaire was developed for the survey and pretested on 50 households from a locality not included in the trial but with similar socio-demographic conditions. The survey was conducted using the Computer Assisted Personal Interview (CAPI) data collection method.

Community mobilization activities were conducted following the baseline survey, and health camps were established. A detailed plan involving UNICEF and relevant partners was developed for community mobilization. Information, education, and communication (IEC) material were developed to address different segments of the target audience, building upon the experience of the community engagement and mobilization strategy from the prior cRCT [20]. Individual sessions with male groups, female groups, and healthcare providers were held in all target areas. Sessions with influencers (community and religious leaders, teachers, and other influencers) were held at the Union Council level.

The community mobilization plan and IEC material consisting of a pictorial booklet and counselling cards were prepared during the inception phase and ratified by all relevant stakeholders. The IEC materials also contained information on maternal health, nutrition, hygiene and sanitation, routine immunization, polio, and health camps. Teams comprising two female and two male community mobilisers were recruited and trained to deliver the information in the IEC material. They were provided with an IEC booklet and counselling cards as job aids. Each team covered four clusters and delivered key messages through individual sessions with parents and group sessions with male groups, female groups, and healthcare providers at the cluster level. Sessions with community and religious leaders, teachers, and other prominent persons were held at the union council level. Typically, 15 to 20 people attended the male and female group sessions, while 10 to 15 attended the other sessions. A limited number of sessions were held with healthcare providers and influencers at the introduction of the project, with a focus on the importance of routine immunization and OPV/IPV administration.

Along with community mobilization, MNCH camps were established with the consensus of the polio emergency operating cell (EOC) and relevant stakeholders. Before the establishment of health camps, announcements were made at the community level and invitation cards and information about health camps were distributed in advance in each target area. Families were encouraged to visit these health camps. The health camps offered MNCH services and routine immunizations plus IPV as needed. Good clinical practice guidelines and safe injection practices were followed, and the cold chain was also strictly monitored. We aimed to vaccinate each child under the age of five years in the target areas.

The camps generally started after the supplementary immunization activities (SIAs) rounds with the consensus of EOC in the targeted areas and lasted 3 days. They were open for 6–7 h per day. They included staff with basic essential drugs and commodities assessed as appropriate for ambulatory care for MCH at the primary care level. In each union council, 2 fixed and 2 mobile health camps were organized per round and 6 rounds of health camps (one round per month) were organized in target areas to improve community access. Each health camp was staffed by a medical officer, one vaccinator, one paramedic, two facilitators, and a union council supervisor. Staff provided counselling on hygiene, nutrition, and routine immunizations and undertook general maternal and child health assessments. The health camp staff were trained to provide standard medical care, follow good clinical practice, maintain a strict cold chain, and follow safe injection practices. The healthcare providers in the camps delivered nutrition interventions such as micronutrient supplements, OPV, IPV, as well as routine immunizations, and ANC including tetanus toxoid to pregnant women. Primary care medications for common illnesses were administered per the WHO and Government of Pakistan guidelines.

### 2.4. Statistical Analysis

All data were double-entered using Visual FoxPro and backed up at the data management unit at Aga Khan University, with hard copies archived at the institution’s data repository. Demographic, clinical, and vaccination data were merged and analyzed using Stata (version 16.0).

We calculated overall and region-specific vaccination coverage at baseline and endline. A mixed-effects linear regression model estimated the percentage point difference in coverage between baseline and endline. The clustered nature of the data was accounted for by including each cluster as a random effect. Estimates were adjusted for survey design and sampling weights by treating the UC or tehsil as strata and the clusters as primary sampling units. The number of doses administered was the count of all doses of OPV, IPV, and EPI vaccines, excluding unused doses and wastage.

A conceptual framework was developed to classify candidate predictors as distal-, intermediate-, and proximal-level predictors of ecological-level changes in immunity (Figure 2). Bivariate and multivariable associations between socio-demographic factors and changes in the proportion of fully vaccinated children at the UC level were assessed using hierarchical linear regression models; backwards variable selection was performed using *p*-value thresholds of <0.2 and <0.1 at the bivariate and multivariable stages, respectively. Survey responses were aggregated as proportions at the UC level. A child is considered fully vaccinated if they have received all age-specific vaccines in accordance with the national EPI schedule.

The trial received approval from the Ethics Review Committee of Aga Khan University, Pakistan [3307-Ped-ERC-14] and the National Bioethics Committee, Pakistan. Consent was obtained from the parents of the children who participated in the study. The trial was registered on ClinicalTrials.gov under the identifier NCT01908114.

## 3. Results

Between 2014 and 2016, the study involved a total of 10,348 clusters, including 4803 at baseline, 885 at midline, and 4660 at endline. At baseline, out of 4803 clusters, 1524 clusters were selected from Karachi, 3256 were selected from KPK, and 23 were selected from Baluchistan; 37,575 children under 5 years of age were recorded in Karachi, 83,667 were recorded in KPK, and 1708 were recorded in Baluchistan. A total of 4660 clusters remained in the study at the endline (1535 clusters in Karachi, 3115 clusters in KP, and 10 clusters in Baluchistan, with 34,912, 97,712 and 1372 children under 5 years of age in Karachi, KPK, and in Baluchistan, respectively (Figure 3).

Sociodemographic characteristics of the enrolled children and their families at baseline were relatively similar in all three areas of Pakistan (Table 1). A male predominance was seen across all study areas. Age distribution demonstrates a statistically significant high proportion of children in the 24–59 months age group among the study areas. The maternal literacy rate was 44.5%, with the highest found in Karachi (48.4%) and the lowest found in Baluchistan (13.1%). Similarly, the families of the two poorest quintiles were found more in Karachi (39.5%) with a relatively lesser proportion in Baluchistan (4.7%). Differences were also observed regarding access to improved sources of drinking water, methods used for cleaning the drinking water, and improved sanitation. More than 60% of people in Karachi and Baluchistan had access to improved drinking water sources. In contrast, methods used for cleaning the drinking water were found less in Baluchistan (24.1%) and KP (14.1%) in comparison to Karachi (40.6%). Moreover, access to improved sanitation was highest in Karachi 95.5%, followed by Baluchistan (70.1%) and KP (51.4%).

The vaccination status of children younger than 5 years at baseline and endline is shown in Appendix A. At baseline, 55.9%, 35.6%, and 26% were found to be fully immunized in Karachi, KP, and Baluchistan, respectively. At endline, this percentage increased to 60% or more in all three study areas. There was a marked increase in coverage of all OPV doses across all three provinces, with more than 70% of children having received four doses of OPV. The proportion of children receiving three doses of pentavalent and PCV also increased substantially. Coverage with two doses of measles vaccine increased to over 50% in all three areas. At baseline, the coverage of IPV was only 5.3% in KP, 17.8% in Karachi, and 4.4% in Baluchistan. These proportions increased to 36.9%, 50.6%, and 73.8%, respectively, at endline.

As shown in Appendix A, despite the gender imbalance in RI coverage at baseline, a statistically significant female predominance was seen among health camp attendees across all three study sites. A statistically significant difference was achieved in the attendance of children aged under 5 in all three provinces with the highest number of children presented by KP (69.1%). A few females received antenatal care (ANC) from the health camps with the lowest rates recorded in Karachi (0.9%). Regarding the availability of vaccine cards, less than a quarter of the participants had shown it from all three study areas, i.e., 21.6% from Baluchistan, 18.4% from KP, and 12.8% from Karachi.

Overall, 12,074 BCG doses were administered in Karachi, 20,562 were administered in KP, and 4015 were administered in Baluchistan at health camps. Other routine immunization doses included 84,060 OPV doses, 25,669 pentavalent doses, 24,867 PCV doses, 79,820 measles doses, and 128,812 IPV doses in Karachi. Likewise, 57,333 OPV doses, 40,329 pentavalent doses, 37,713 PCV doses, 39,281 measles doses, and 83,160 IPV doses were recorded in KP. In Baluchistan, 5891 OPV doses, 6841 pentavalent doses, 6807 PCV doses, 20,159 measles doses, and 8466 IPV doses were recorded. In contrast, refusals for routine immunization and OPV were reportedly highest in Karachi (1% and 0.1%) and lowest in KP (<0.1% and <0.1%), respectively. The greatest proportion for IPV refusal was seen in KP (14%). Such differences among study areas were statistically significant.

Across the study sites, the crude estimates of routine immunization coverage at baseline and endline for children under 5 years were also evaluated, and statistically significant differences were obtained. Appendix A reflects that in all, 41.6% (95% CI 41.4–41.9) and 62.3% (95% CI 62–62.5) of children aged under 5 were fully immunized at baseline and endline, respectively, with a significant raise in coverage of 20.6% (95% CI 20.2–21). Region-wise-estimated crude changes are also demonstrated in Appendix A**.**

Information about IPV coverage shows an overall greater proportion of children vaccinated at baseline, i.e., 10.4% to 46.2%, at endline with a significant difference of 35.8% (95% CI 35.4–36.1). Furthermore, coverage of OPV vaccines was quite similar for all doses, i.e., 12.6%, 14.5%, 12.6%, and 12.5%, respectively. A significant raise in the receipt of pentavalent, PCV, and measles vaccines was also seen. PCV and pentavalent coverage of dose 1 was relatively higher with a difference of 26.7% (95% CI 26.4; 27.1) and 23.8% (95% CI 23.5; 24.2), respectively, from endline to baseline as compared to other doses.

Crude estimates for regional distribution pointed towards the increased rates of coverage in Baluchistan from baseline at 26% (95% CI 24; 28.1) to endline at 60.4% (95% CI 31.1; 37.7) with a greater difference of 34.4% (95% CI 31.1; 37.7) when compared to other regions. Immunization coverage of Karachi from baseline at 55.9% (95% CI 55.4; 56.4) to at 67.6% (95% CI 67.1; 68.1) was noted with a significant difference of 11.7% (95% CI 11; 12.4), which was relatively very low.

Similarly, regional distribution of full immunization coverage revealed a significant difference of 11.7% in Karachi, 24.8% in KP, and 34.4% in Baluchistan from baseline to endline. The most evident uptake of routine immunization including OPV, PCV, pentavalent, and measles was seen in Baluchistan between the study period. Furthermore, the IPV coverage was somewhat similar in all three study sites but was comparatively high in Baluchistan from baseline at 34.4% (95% CI 32.2; 36.7) to endline at 73.8% (95% CI 71.4; 76.1) with a difference of 39.5% (95% CI 36.2; 42.7), followed by KP and Karachi as shown in Appendix A.

Figure 4 describes the crude estimates of the mean rise in immunization coverage across the study sites during baseline and endline. The average update in Karachi was between 55–70%, 35–65% in KP, and 25–60% in Baluchistan. The maximum increment in vaccine acceptance was noticed in Baluchistan.

In Table 2 and Table 3, full immunization rates are assessed on three levels of the framework: the distal, intermediate (access and environment), and proximal level (camp attendance and effectiveness) using bivariate and multivariate analyses. At the distal level, on multivariate analysis, among various indicators, family size was found to be a significant predictor of change in immunity in families with a family size of more than six members; their full immunization was significantly better (β = 0.68; *p* = <0.0001). Bivariate analysis reflected that the likelihood of full immunization coverage increased with decreased Pashto speaking (β = 0.1; *p* = 0.045). Province wise, KP and Baluchistan have improved coverage compared to Karachi (β = 0.15; *p* = 0.003).

At the intermediate level, the likelihood of full immunization decreased with the decrease in knowledge about vaccination (β = −0.38; *p* = 0.002), knowledge about polio vaccine (β = −0.25; *p* = 0.011), and knowledge about IPV (β = −0.06; *p* = 0.546). Perceived obstacles to vaccination were fear of adverse events (β = −0.4; *p* = <0.0001) and lack of education (β = 0.23; *p*= 0.031), which were found to be significant during bivariate and multivariate analyses. Moreover, partial immunization significantly increased the likelihood of full immunization, which reflected that the probability of full immunization would also increase in every single unit of partial immunity.

At proximal level, community mobilization (β = 0.26; *p* = 0.008) and attendance to health camp (β = 0.21; *p* = <0.0001) was found to enhance full immunization coverage. On the other hand, the most prominent reason for not attending health included no need to attend the health camp as the child was not ill (β = −0.13; *p* = 0.008). Regarding the attendance and effectiveness of the health camps, significant factors included female gender, children aged under 5, and receipt of routine immunization.

## 4. Discussion

The present evaluation demonstrated a substantial increase in the coverage of RI along with polio vaccines throughout the project. Despite uncertain security conditions, population restrictions, and vaccine hesitancy, the favorable combined impact of community mobilization, and delivery of MNCH and immunization services through temporary MCH-focused health camps during SIAs was evident. Furthermore, this intervention was effective in increasing vaccine coverage and considered feasible and acceptable by the community. Approximately 480,762 children aged under 5 were vaccinated with more than 200,000 IPV doses, while 16,539 females received antenatal care services with around 70% OPV coverage and 60% full immunization with an outstanding safety profile. In contrast to the negative concerns regarding the uptake and administration of polio vaccines, the highest proportion of IPV coverage was seen in Baluchistan (73.8%), despite it being a traditional high-risk zone with a historically poor performance in terms of the polio program. This enhancement in OPV and IPV coverage is promising in achieving the threshold required for stopping poliovirus circulation.

Childhood immunizations have had a checkered past in Pakistan. Along with other social determinants, poor access and variable services have contributed to poor child coverage of RIs [23]. One of this study’s strengths was the community’s unequivocal participation and acceptance of this approach. This also reflected the widespread unmet need for maternal and child health and immunization services in these areas. Notwithstanding this overall success, our study revealed certain limitations in the program’s execution. Due to the high demand for these services at the health camps, at times it became difficult for the staff to control the incoming population and so local community leaders and volunteers were approached to maintain order. However, no major adverse events were encountered during the vaccination process. Vaccine hesitancy remained an issue and needed repeated community outreach given the widely reported adverse effects in some parts of Khyber Pakhtunkhwa following a mass measles vaccination campaign [24]. Therefore, experienced vaccinators were hired from the public sector and were given extensive training on WHO standards of safe injection practice [25].

In addition to this, storage of vaccine and its administration needed extra attention given the hot weather conditions. Sufficient ice packs, cold storage boxes, and back-up thermometers had to be readily available. Furthermore, staff were trained to observe vaccine vial monitors before administering the vaccines. Accessing high-risk areas and convincing chronic vaccine refusals was also a significant strength of the study, although achieving universal coverage and access remained a challenge [26] as some areas were under local curfews during the conduct of the study, and staff could only return to those sites as and when possible. The present study is one of its kind to reach at-risk populations in insecure and high-risk areas of Karachi, KP, and Baluchistan.

Our study findings support the assumption that polio vaccines can be well received if they are delivered as part of a package of health services, and this could be a better way to engage local communities, religious leaders, and other stakeholders rather than polio-specific programs [22]. On a comparable note, Nigeria demonstrated the viability of a successful campaign with IPV immunization in its conflict areas [27]. However, linking such short-term camps to SIAs invites more costs and the additional requirement of human resources, experienced personnel, trainings, monitoring, and evaluation. On a similar note, Kenya also followed this pragmatic approach. It provided OPV and IPV in temporary health facility sites. At the same time, door-to-door mobilization of caregivers was carried out to encourage them to bring their children to the vaccination sites. They achieved a rapid coverage of vaccines in refugee populations at risk of outbreaks [28]. The key learnings from our study have implications for reaching refugee and migrant populations in conflict-affected and insecure settings. Either fixed or mobile, health camps with community mobilization and advocacy strategies focused on promoting general MNCH and immunizations can be more effective in such circumstances. Such approaches were also used in Angola [29], Afghanistan [30], India [31], Jordan, and Lebanon [32] and have also been used among refugee children in Europe [33] affected by conflict and uncertainty and natural calamities. Similarly, a study in the Central African Republic showed that vaccination coverage for children aged 6 weeks to 59 months could be increased following a preventive, multi-antigen vaccination campaign in conflict areas [34]. The current study provides strong evidence that trust for OPV and IPV can be built despite mistrust and insecurity, provided one uses an integrated approach and engages the community.

Polio-eradication efforts in Pakistan continue to suffer ongoing challenges; yet, the situation is much improved, and despite the challenge of COVID-19-related interruption, the program has shown resilience. The areas with continuous poliovirus circulation in Quetta, FATA, and upper Sindh are sometimes inaccessible, despite the country striving to eliminate poliovirus. Our findings have paved the way for policymakers by highlighting the influence of community mobilization and the establishment of health camps to increase vaccine coverage rates and has been widely emulated by the program in the insecure areas of KPK and Baluchistan. These encouraging results from Pakistan can address the residual global challenges of residual parts where polio vaccination is affected by geographical, political, or war-related insecurities. [6,35]. At this level, the polio program in Pakistan, as part of its National Emergency Action Plan 2021–2023, has realigned its polio SIAs approach. The emphasis now extends beyond conventional methods, prioritizing vulnerable subpopulations and chronically missed children. This shift involves the targeted use of effective interventions like integrated health camps in the polio high-risk district across the country. These health camps operate on a rotational basis, occurring after each national immunization campaign day, and provide routine vaccinations including IPV and OPV and preventive maternal and child health services [36].

Nonetheless, our project had some challenges in its execution. At times, the high demand for services at the health camps made it difficult for project staff to control the crowds that turned out. To address this issue, we involved the community leaders and local community volunteers to help maintain order. Another challenge was the maintenance of the vaccine cold chain in camps. To address this, we provided adequate volumes of ice packs and backup thermometers to the vaccination teams. Additionally, the vaccines used in the study had vaccine vial monitors and staff underwent thorough training in their use. The study is also limited by the fact that coverage estimates were largely based on family reports and doses administered at the campsite. This reliance on family reporting could introduce potential biases in the coverage estimates and is likely the only source of information in such conflict-affected insecure areas. Future approaches could also rely on vaccination records, paper or electronic.

## Figures and Tables

**Figure 1 vaccines-12-00089-f001:**
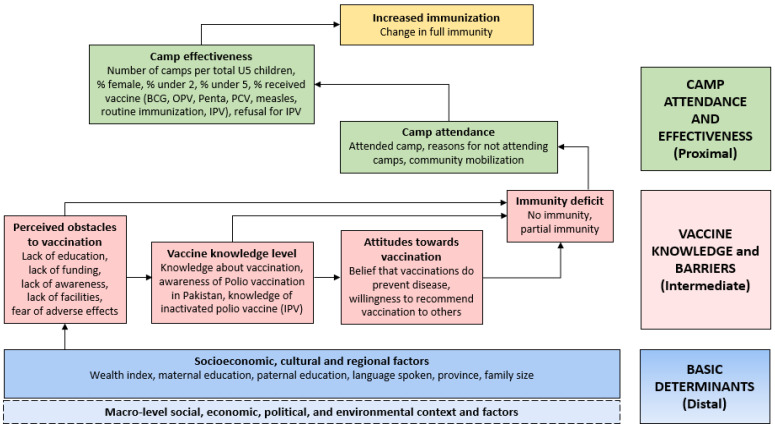
Conceptual model for determinants of increased immunization at UC level.

**Figure 2 vaccines-12-00089-f002:**
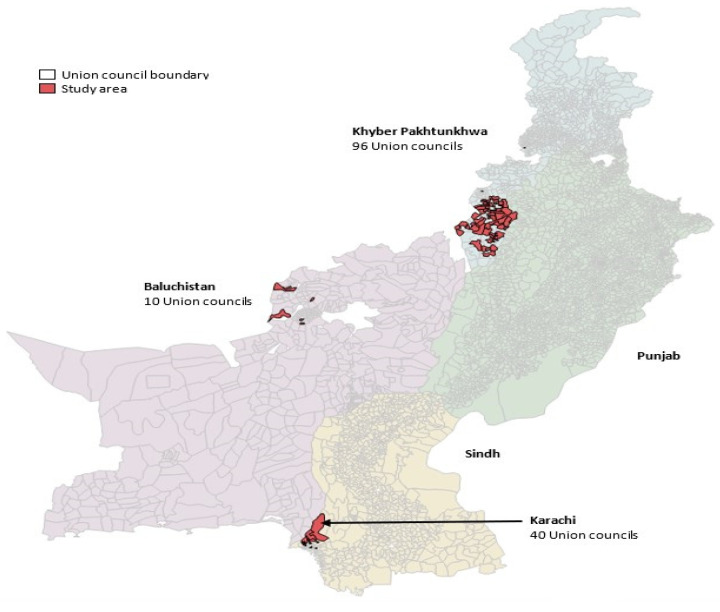
Map of Pakistan showing study areas.

**Figure 3 vaccines-12-00089-f003:**
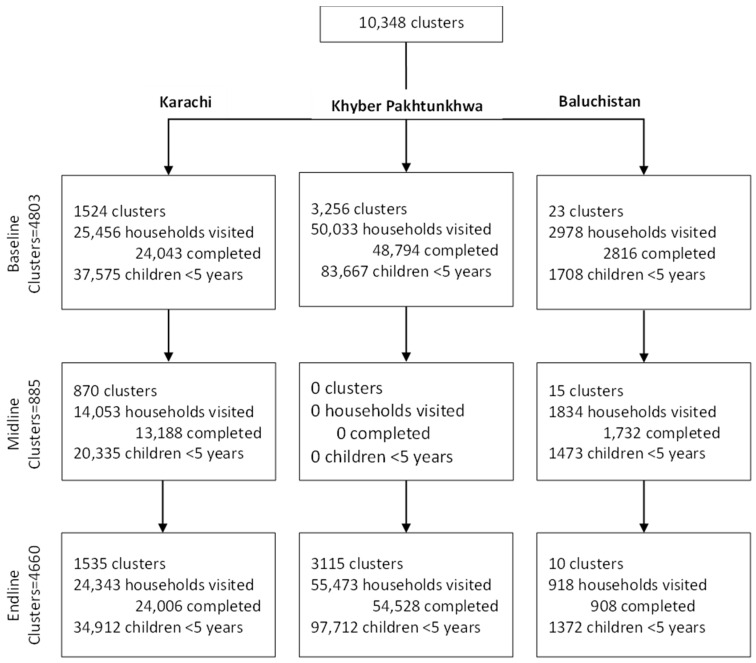
Study profile.

**Figure 4 vaccines-12-00089-f004:**
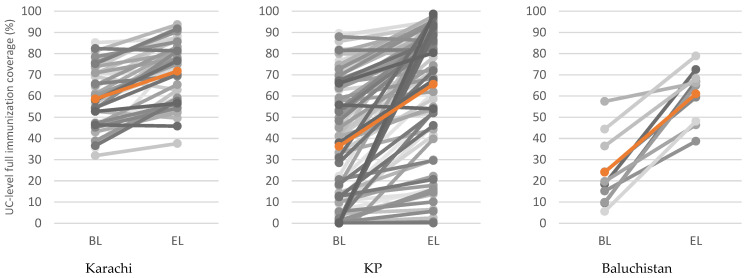
Estimated change in crude UC-level full immunization coverage (%) between baseline (BL) and endline (EL) by region (mean change highlighted and represented by the orange lines).

**Table 1 vaccines-12-00089-t001:** Baseline socio-demographic characteristics of children younger than 5 years and their families (*n* = 122,950).

	Overall	Karachi	KP	Baluchistan	*p*-Value
Children < 5 years old at Baseline	122,950	37,575	83,667	1708	
Female	55,470 (45.1%)	18,292 (48.7%)	36,359 (43.5%)	819 (48.0%)	<0.001 *
Age (months)					
0–5	8355 (6.8%)	4130 (11.0%)	4050 (4.8%)	175 (10.3%)	<0.001 *^,‡^
6–11	10,734 (8.7%)	3828 (10.2%)	6723 (8.0%)	183 (10.7%)	<0.001 *^,‡^
12–23	25,970 (21.1%)	7677 (20.4%)	17,947 (21.5%)	346 (20.3%)	<0.001 *
24–59	77,891 (63.4%)	21,940 (58.4%)	54,947 (65.7%)	1004 (58.8%)	<0.001 *^,‡^
Maternal literacy	54,733 (44.5%)	18,203 (48.4%)	36,307 (43.4%)	223 (13.1%)	<0.001 *^,†,‡^
Wealth index—poorest 2 quintiles	47,281 (38.5%)	14,823 (39.5%)	32,378 (38.7%)	80 (4.7%)	<0.001
Improved source of drinking water	41,881 (34.1%)	23,520 (62.6%)	17,306 (20.7%)	1055 (61.8%)	<0.001 *^,‡^
Method for cleaning drinking water	27,489 (22.4%)	15,259 (40.6%)	11,819 (14.1%)	411 (24.1%)	<0.001 *^,†,‡^
Improved sanitation	80,078 (65.1%)	35,874 (95.5%)	43,006 (51.4%)	1198 (70.1%)	<0.001 *^,†,‡^
Improved sanitation	80,078 (65.1%)	35,874 (95.5%)	43,006 (51.4%)	1198 (70.1%)	<0.001 *^,†,‡^

Abbreviations: EPI = Expanded Program on Immunization, KP = Khyber Pakhtunkhwa, OPV = Oral Polio vaccine * *p* < 0.001 (Bonferroni corrected) for Karachi vs. KP; ^†^ *p* < 0.001 (Bonferroni corrected) for Karachi vs. Baluchistan; ^‡^ *p* < 0.001 (Bonferroni corrected) for KP vs. Baluchistan.

**Table 2 vaccines-12-00089-t002:** Bivariate regression results for change in full immunity among children aged under 5 at the UC level (*n* = 141).

Outcome: Change in Full Immunity (%)	*n* = 141 UCs		
	Univariate		
Characteristic	β	95% CI	*p*-Value
**Distal level**			
**Socioeconomic, cultural and regional**			
Wealth quintile (richest 2 vs. poorest 3)	−0.08	(−0.26; 0.11)	0.410
Maternal literacy (%)	−0.03	(−0.23; 0.17)	0.773
Paternal literacy (%)	−0.23	(−0.47; 0.01)	0.059
Pashto-speaking (20−80% vs. <20%)	−0.10	(−0.21; 0.01)	0.075
Pashto-speaking (20–80% vs. >80%)	0.10	(0.002; 0.19)	0.045
Province (KP vs. KHI)	0.15	(0.05; 0.24)	0.003
Province (BAL vs. KHI)	0.24	(0.06; 0.41)	0.008
Family size >6 (%)	0.68	(0.49; 0.88)	<0.001
**Intermediate level (Access and environment)**			
**Vaccine knowledge level**			
Know about vaccination (%)	−0.38	(−0.63; −0.14)	0.002
Know about Polio vaccine in Pakistan (%)	−0.25	(−0.44; −0.06)	0.011
Know about IPV (%)	−0.06	(−0.24; 0.13)	0.546
**Perceived obstacles to vaccination**			
Lack of education (%)	0.38	(0.14; 0.62)	0.002
Lack of funding (%)	0.42	(0.05; 0.78)	0.025
Lack of awareness (%)	0.16	(−0.11; 0.43)	0.254
Lack of facilities (%)	−0.05	(−0.28; 0.17)	0.636
Fear of adverse effects (%)	−0.40	(−0.58; −0.22)	<0.001
**Personal barriers to vaccination**			
Will recommend vaccination to others (%) (80+% vs. <80%)	−0.07	(−0.16; 0.02)	0.129
Think vaccinations prevent disease (%)	−0.41	(−0.65; −0.16)	0.001
**Immunity level prior to camps**			
Partial immunity (%)	0.46	(0.25; 0.68)	<0.001
No immunity (10–50% vs. <10%)	0.09	(0; 0.18)	0.061
No immunity (50%+ vs. 10–50%)	0.17	(0.04; 0.3)	0.013
**Proximal level (Camp attendance and effectiveness)**			
**Camp attendance**			
Primary source of information about camps was community mobilization (%)	0.26	(0.07; 0.45)	0.008
Attended camp (80+% vs. <80%)	0.21	(0.13; 0.29)	<0.001
**Reason for not attending health camps**			
Not needed as child was not ill (any % vs. 0%)	−0.29	(−0.38; −0.19)	<0.001
It was too far away (any % vs. 0%)	−0.26	(−0.36; −0.16)	<0.001
Family did not allow (any % vs. 0%)	−0.26	(−0.36; −0.16)	<0.001
No camps (any % vs. 0%)	−0.15	(−0.24; −0.07)	0.001
**Camp effectiveness**			
Number of camps per total under 5 population	−0.01	(−0.02; 0.01)	
Female (%)	−1.04	(−1.7; −0.37)	0.003
Visitors under 2 (%)	0.07	(−0.2; 0.34)	0.606
Visitors under 5 (%)	0.59	(0.29; 0.89)	<0.001
Received BCG vaccine (%)	0.18	(0.09; 0.27)	<0.001
Received OPV (%)	0.42	(0.21; 0.62)	<0.001
Received Penta vaccine (%)	0.38	(0.12; 0.64)	0.005
Received PCV (%)	0.39	(0.03; 0.75)	0.035
Received measles vaccine (%)	1.06	(0.79; 1.32)	<0.001
Received routine immunization (under 5) (%)	0.73	(0.48; 0.97)	<0.001
Received routine immunization (zero dose) (%)	0.07	(0.04; 0.1)	<0.001
Received IPV (4–23 months) (%)	0.23	(0.01; 0.45)	0.043
Refused for IPV (1–20% vs. <1%)	0.21	(0.11; 0.3)	<0.001
Refused for IPV (20%+ vs. <1%)	0.06	(−0.04; 0.17)	0.236

Abbreviations: Bacillus Calmette–Guérin vaccine, IPV = Inactivated Polio vaccine, OPV = Oral Polio vaccine, PCV = Pneumococcal conjugate vaccine.

**Table 3 vaccines-12-00089-t003:** Multivariate hierarchical regression results for change in full immunity among children aged under 5 at the UC level (*n* = 141).

Outcome: Change in Full Immunity (%)			
	Multivariate		
Characteristic	β	95% CI	*p*-Value
**Distal level**			
**Socioeconomic, cultural, and regional**			
Family size > 6 (%)	0.68	(0.49; 0.88)	<0.001
**Intermediate level (access and environment)**			
**Perceived obstacles to vaccination**			
Lack of education (%)	0.23	(0.02; 0.43)	0.031
**Immunity level prior to camps**			
Partial immunity (%)	0.32	(0.12; 0.51)	0.002
**Proximal level (camp attendance and effectiveness)**			
**Camp attendance**			
Primary source of information about camps was community mobilization (%)	0.14	(−0.01; 0.29)	0.068
**Reason for not attending health camps**			
Not needed as child was not ill (any % vs. 0%)	−0.13	(−0.22; −0.03)	0.008
**Camp effectiveness**			
Female (%)	1.72	(0.88; 2.56)	<0.001
Visitors under 5 (%)	0.93	(0.54; 1.33)	<0.001
Received BCG vaccine (%)	0.08	(−0.01; 0.18)	0.086
Received OPV (%)	0.31	(0.03; 0.59)	0.031
Received PCV (%)	−0.91	(−1.37; −0.46)	<0.001
Received measles vaccine (%)	0.59	(0.16; 1.01)	0.007
Received routine immunization (zero dose) (%)	−0.04	(−0.08; 0.001)	0.057

Abbreviations: Bacillus Calmette–Guérin vaccine, OPV = Oral Polio vaccine, PCV = Pneumococcal conjugate vaccine.

## Data Availability

The data presented in this study are available on request from the corresponding author. The data are not publicly available due to privacy and ethical concerns.

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
