# Peer review of "A Holistic Strategy of Mother and Child Health Care to Improve the Coverage of Routine and Polio Immunization in Pakistan: Results from a Demonstration Project"

_vaccines, 2024, doi:10.3390/vaccines12010089_

Round 1
Reviewer 1 Report
Comments and Suggestions for Authors
While the article is well-written, it requires some enhancements for final acceptance. I have a few suggestions such as:
Introduction
The introduction should be enhanced by referencing previous research on the factors influencing Routine and Polio Immunization utilization in Pakistan. In its current state, the authors overlook mentioning any past studies on childhood vaccine utilization, leaving readers and reviewers unconvinced about the study's rationale. To address this, I recommend strengthening the introduction by highlighting routine immunization coverage in South Asia, with a specific focus on Pakistan. Clearly outline the existing research gaps and articulate how this study aims to contribute to policy and practice by addressing these gaps.
Method
The method section lacks clarity as it does not specify the socio-demographic factors considered, nor does it detail the measurement criteria for a "fully vaccinated child." To enhance understanding for readers, it is suggested that the authors provide elucidation on these aspects to ensure a more comprehensive and transparent presentation of the research methodology.
Results
The typo in the second line (i.e. “.”) should be corrected.
Regarding Table 5, which pertains to the Multivariate hierarchical regression results for the change in full immunity among children under 5 at the UC level, authors are suggested to include the 95% Confidence Interval for clarity. The presentation of bivariate and multivariate results in the same table is unclear, the Table 5 is difficult to comprehend. Please provide clarification or reorganize the table to enhance its comprehensibility.
Discussion
The discussion section should delve into the policy implications rather than merely stating that "Our findings have paved the way for policymakers by highlighting the influence of community mobilization and the establishment of health camps to increase vaccine coverage rates, which has been widely emulated by the program in the insecure areas of KPK and Baluchistan." Please provide a more detailed analysis and discussion of the specific policy implications derived from the study's findings.
Please elaborate on the novelty of these findings and how the current study can be applied to enhance Mother and Child Health Care, particularly in improving the coverage of Routine and Polio Immunization in Pakistan. Additionally, provide insights into why this study is considered a "Holistic Strategy" compared to other studies, offering a more comprehensive perspective.
The study does not address the "limitations" associated with the research.
Author Response
Reviewer 1:
Comments and Suggestions for Authors
While the article is well-written, it requires some enhancements for final acceptance. I have a few suggestions such as:
Introduction
Comment: The introduction should be enhanced by referencing previous research on the factors influencing Routine and Polio Immunization utilization in Pakistan. In its current state, the authors overlook mentioning any past studies on childhood vaccine utilization, leaving readers and reviewers unconvinced about the study's rationale. To address this, I recommend strengthening the introduction by highlighting routine immunization coverage in South Asia, with a specific focus on Pakistan. Clearly outline the existing research gaps and articulate how this study aims to contribute to policy and practice by addressing these gaps.
Response: Thank you for the comment. We have added the following paragraph discussing the RI coverage in South Asian countries.
“In the South Asian region, an estimated 3 million children under one year of age remain unvaccinated, according to the 2022 estimates from the WHO and UNICEF [18]. Bangladesh stands out with the highest proportion of children receiving complete childhood immunization (88.2%), followed by Nepal (79.2%), India (77.1%), and Pakistan (76.5%) [19,20]. On the other hand, Afghanistan reports the lowest immunization rates at 42.6% [19]. Notably, when considering POV3, all South Asian countries reported to have coverage exceeding 90%, except for Afghanistan, Nepal, and Pakistan [18].”
For the last part of the question our paper explains the background information, and gap that this study has fulfilled lines 113 to 135 which was to build trust and demand for polio vaccines requires through effective communication and community mobilization by implementing a community mobilization strategy and community-based maternal and child health camps including administration of RI and IPV through cluster randomized controlled trial (cRCT) in several insecure areas of the country.
Method
Comment: The method section lacks clarity as it does not specify the socio-demographic factors considered, nor does it detail the measurement criteria for a "fully vaccinated child." To enhance understanding for readers, it is suggested that the authors provide elucidation on these aspects to ensure a more comprehensive and transparent presentation of the research methodology.
Response: We appreciate the valuable comment from the reviewer. In the methods section under the subheading ‘Statistical analysis’ the paper has mentioned that socio-demographic factors are considered in estimating the regression model. Further to that we have added the following definition of fully vaccinated in the revised paper. “A child is considered fully vaccinated s/he has received all age-specific vaccines in accordance with the national EPI schedule”.
Results
Comment: The typo in the second line (i.e. “.”) should be corrected.
Response: Thank you for highlighting this, we have rectified the typo.
Comment: Regarding Table 5, which pertains to the Multivariate hierarchical regression results for the change in full immunity among children under 5 at the UC level, authors are suggested to include the 95% Confidence Interval for clarity. The presentation of bivariate and multivariate results in the same table is unclear, the Table 5 is difficult to comprehend. Please provide clarification or reorganize the table to enhance its comprehensibility.
Response: Thank you for providing your valuable feedback. We have incorporated your suggestion by organizing the bivariate and multivariate analyses into distinct tables. Specifically, we have created Table 2 for the bivariate analysis with confidence intervals (CIs) and Table 3 for the multivariate analysis with CIs. We believe this modification will significantly improve the clarity and comprehensibility of the information presented.
Discussion
Comment: The discussion section should delve into the policy implications rather than merely stating that "Our findings have paved the way for policymakers by highlighting the influence of community mobilization and the establishment of health camps to increase vaccine coverage rates, which has been widely emulated by the program in the insecure areas of KPK and Baluchistan." Please provide a more detailed analysis and discussion of the specific policy implications derived from the study's findings.
Response: Thank you for the comment. We have added the following paragraph to the discussion section, which talks about the policy implications of the findings from this study.
“At this level, the polio program in Pakistan, as part of its National Emergency Action Plan 2021-2023, has realigned its polio SIAs approach. The emphasis now extends beyond conventional methods, prioritizing vulnerable subpopulations and chronically missed children. This shift involves the targeted use of effective interventions like integrated health camps in the polio high-risk district across the country. These health camps operate on a rotational basis, occurring after each national immunization campaign day, and provide routine vaccinations including IPV and OPV and preventive maternal and child health services [35].”
Comment: Please elaborate on the novelty of these findings and how the current study can be applied to enhance Mother and Child Health Care, particularly in improving the coverage of Routine and Polio Immunization in Pakistan. Additionally, provide insights into why this study is considered a "Holistic Strategy" compared to other studies, offering a more comprehensive perspective.
Response: This is the second study that was scientifically designed to evaluate a community-based strategy for reaching high-risk populations in conflict-affected and insecure areas. The strategy involved delivering an integrated package of maternal and child health along with immunization services through health camps, complemented by community mobilization efforts. The results indicate that community mobilization and attendance at health camps significantly enhanced full immunization coverage, underscoring the importance of community engagement and targeted interventions in improving immunization rates. By the endline, full immunization coverage had increased to 60% or more in all three study areas compared to the baseline. Additionally, there was a significant increase in the coverage of both OPV and IPV across all three provinces at the endline.
Comment: The study does not address the "limitations" associated with the research.
Response: Thank you for pointing this out. We have added the following paragraph to the revised manuscript.
“Our study had some challenges in execution. At times, the high demand for services at the health camps made it difficult for project staff to control the crowds that turned out. To address this issue, we involved the community leaders and local community volunteers to help maintain order. Another challenge was the maintenance of the vaccine cold chain in camps. To address this, we provided adequate volumes of ice packs and backup thermometers to the vaccination teams. Additionally, the vaccines used in the study had vaccine vial monitors and staff underwent thorough training in their use. The study was limited by the fact that coverage estimates were largely based on family reports and doses administered at the campsite were the primary basis for these estimates. This reliance on family reporting may have introduced potential biases and limitations to the coverage estimates.”
Reviewer 2 Report
Comments and Suggestions for Authors
1)ABSTRACT In the abstract, the statistical technique must be mentioned in the material and methods section. The results say that a multivariable technique was carried out, and a beta coefficient is indicated. Still, it is impossible to know if it refers to a logistic regression, a multiple linear regression, etc., which is why it is tough to interpret.
2) INTRODUCTION
- The mention of the CIA-guided campaign could be perceived as politically charged. Maintaining a neutral tone and focusing on the scientific aspects is important. You might consider rephrasing to focus on how external factors, including political or intelligence campaigns, have impacted public trust in immunization programs. (Please keep the references)
MATERIAL AND METHODS
Regarding the specification of p-value thresholds (<0.2 and <0.1) at different stages of analysis (bivariate and multivariable) justify the choice of these specific thresholds, as they are somewhat unconventional (common thresholds are usually 0.05 or 0.01).
Authors mention using log, square root, and cubic transformations for skewed distributions. Explain how the decision was made to use these specific transformations. For example, do you use skewness, kurtosis, or some test like Wilkinson Shapiro? Also, when using logs, what do you do with cero values, did you use Tukeys method. Log (value + (1% Mean))?
DISCUSSION
Change "bounced back" (line 419) into "has shown resilience." Because it too colloquial for a scientific paper.
Author Response
Reviewer 2:
Comments and Suggestions for Authors
Comment: 1) ABSTRACT In the abstract, the statistical technique must be mentioned in the material and methods section. The results say that a multivariable technique was carried out, and a beta coefficient is indicated. Still, it is impossible to know if it refers to a logistic regression, a multiple linear regression, etc., which is why it is tough to interpret.
Response: Thank you for your thorough review. We have added the following sentence describing the statistical technique to the method section.
“Multivariable associations between socio-demographic factors and changes in the proportion of fully vaccinated children at the UC level were assessed using hierarchical linear regression models”.
Comment: 2) INTRODUCTION
- The mention of the CIA-guided campaign could be perceived as politically charged. Maintaining a neutral tone and focusing on the scientific aspects is important. You might consider rephrasing to focus on how external factors, including political or intelligence campaigns, have impacted public trust in immunization programs. (Please keep the references)
Response: We revised the sentence as suggested to make the tone neutral.
MATERIAL AND METHODS
Comment: Regarding the specification of p-value thresholds (<0.2 and <0.1) at different stages of analysis (bivariate and multivariable) justify the choice of these specific thresholds, as they are somewhat unconventional (common thresholds are usually 0.05 or 0.01).
Response: Thank you again for a thorough review. We agree with the reviewer that the conventional criteria for the p-value is 0.05, but this is for the final model. However, we considered a list of variables significant at bivariate and we considered 0.20 to select a variable for the multivariable model and final model was built with a criteria of p<0.1.
Comment: Authors mention using log, square root, and cubic transformations for skewed distributions. Explain how the decision was made to use these specific transformations. For example, do you use skewness, kurtosis, or some test like Wilkinson Shapiro? Also, when using logs, what do you do with cero values, did you use Tukeys method. Log (value + (1% Mean))?
Response: Thank you for highlighting this crucial point. We sincerely apologize for the oversight, as this element should not have been included in the text. It originated as a consideration during the early stages of data analysis, but subsequent revisions led to its removal. We have now eliminated this sentence from the manuscript to ensure consistency and coherence.
DISCUSSION
Comment: Change "bounced back" (line 419) into "has shown resilience." Because it too colloquial for a scientific paper.
Response: Thank you for the kind suggestion. We have changed the sentence ads suggested.
Reviewer 3 Report
Comments and Suggestions for Authors
I was invited to revise the paper entitled "A Holistic Strategy of Mother and Child Health Care to 2 Improve the Coverage of Routine and Polio Immunization in 3 Pakistan: Results from a Demonstration Project". It aimed to evaluate the impact of an integrated strategy implemented to improve community engagementtowards immunization campaigns in a polio-endemic districts of Pakistan. The topic is relevant for public health facing a crucial point in polio-eradication campaign.
THe paper is well written and the methodology was strong and properly applied.
Observations:
- Introduction should better describe vaccination schedule proposed in Pakistan;
- Among Methods, Authors should better describe the type of intervention performed;
- Discussion section was poor. Authors should compare their results with previous similar study performed also outside Pakistan. Strenght and limitation section should be better added;
- Figure 4 should be better described.
Author Response
Reviewer 3:
Comments and Suggestions for Authors
Comment: I was invited to revise the paper entitled "A Holistic Strategy of Mother and Child Health Care to 2 Improve the Coverage of Routine and Polio Immunization in 3 Pakistan: Results from a Demonstration Project". It aimed to evaluate the impact of an integrated strategy implemented to improve community engagement towards immunization campaigns in a polio-endemic districts of Pakistan. The topic is relevant for public health facing a crucial point in polio-eradication campaign.
The paper is well written and the methodology was strong and properly applied.
Response: thank you positive feedback.
Observations:
Comment: - Introduction should better describe vaccination schedule proposed in Pakistan;
Response: Thank you for the comment, we added a description of the national vaccination schedule to the introduction.
Comment: - Among Methods, Authors should better describe the type of intervention performed;
Response: Thank you for a thorough review. The subheading “procedure” under Methods section describes the type of intervention given – which involved community mobilization and the establishment of Maternal, Newborn, and Child Health (MNCH) camps in targeted areas.
Comment: - Discussion section was poor. Authors should compare their results with previous similar study performed also outside Pakistan. Strength and limitation section should be better added;
Response: Thank you for the feedback. We have added some relevant comparisons to the discussion section as suggested. For study limitations, we added the following paragraph to the revised manuscript.
“Our study had some challenges in execution. At times, the high demand for services at the health camps made it difficult for project staff to control the crowds that turned out. To address this issue, we involved the community leaders and local community volunteers to help maintain order. Another challenge was the maintenance of the vaccine cold chain in camps. To address this, we provided adequate volumes of ice packs and backup thermometers to the vaccination teams. Additionally, the vaccines used in the study had vaccine vial monitors and staff underwent thorough training in their use. The study was limited by the fact that coverage estimates were largely based on family reports and doses administered at the campsite were the primary basis for these estimates. This reliance on family reporting may have introduced potential biases and limitations to the coverage estimates.”
Response: - Figure 4 should be better described.
Response: Thank you for your comment. Figure 4 illustrates the estimated change in crude UC-level full immunization coverage (%) between Baseline (BL) and Endline (EL) in all three regions (Karachi, KP, and Baluchistan). A detailed description of this figure has already been included in the manuscript.
Reviewer 4 Report
Comments and Suggestions for Authors
In this study, Habib et al. deal with a very interesting topic, that is the vaccine acceptance in Pakistan, specifically targeting a subset of the pakistani population resident in 146 districts where high vaccine refusal rates have been previously reported. The study implemented an integrated strategy that specifically attempted to enhance community commitment and engagement in accepting vaccinations. The study was performed in three times (see Figure 3).
The study is both interesting and well documented, but it could benefit from some improvements.
1) The results section is overcrowded with tables and figures that are only discussed in the later sections of the paper. The study could benefit from reducing the tables within the results sections that could be moved to annex / supplementary material (e.g. Table 1b, Table 2-4).
2) Authors should more properly define the assessed outcomes: according to the earlier sections of the paper, Authors seemly focused on the vaccination rates for polio and "other routine vaccinations"; assessed vaccines are discussed across the results and discussion. These section could benefit from a more systematic approach, i.e. reporting separately the various vaccinations in subhedings of the results: 3.1 Polio, 3.2 Measles, and so on.
3) The design of the various clusters is quite heterogenous, at least according to Figure 3. Could you explain these differences?
4) I did expect Table 5 and its content as the core of this study; not only a specific subheading should be therefore identified, but the content of this table should be more extensively reported and discussed.
5) Discussion could benefit from reporting of similar experience in other low income countries where vaccination rates were difficult to achieve because of the difficult acceptance by some population groups.
Comments on the Quality of English LanguageTo be honest, the overall quality of the English text is more than appropriate and acceptable for publication. On the contrary, the text is affected by some typos, particularly in results section (e.g. rows 231 to 237 seem affected by some deletions and repetition, please double check).
Author Response
Reviewer 4:
Comments and Suggestions for Authors
In this study, Habib et al. deal with a very interesting topic, that is the vaccine acceptance in Pakistan, specifically targeting a subset of the pakistani population resident in 146 districts where high vaccine refusal rates have been previously reported. The study implemented an integrated strategy that specifically attempted to enhance community commitment and engagement in accepting vaccinations. The study was performed in three times (see Figure 3).
The study is both interesting and well documented, but it could benefit from some improvements.
Comment: The results section is overcrowded with tables and figures that are only discussed in the later sections of the paper. The study could benefit from reducing the tables within the results sections that could be moved to annex / supplementary material (e.g. Table 1b, Table 2-4).
Response: Thank you for the comment. We acknowledge that the results section was somewhat crowded with large tables, so we have relocated the suggested tables to the supplementary materials for better clarity.
Comment: Authors should more properly define the assessed outcomes: according to the earlier sections of the paper, Authors seemly focused on the vaccination rates for polio and "other routine vaccinations"; assessed vaccines are discussed across the results and discussion. These section could benefit from a more systematic approach, i.e. reporting separately the various vaccinations in subheadings of the results: 3.1 Polio, 3.2 Measles, and so on.
Response: Thank you for your comment. Considering the intricacies of the results and their overlaps, we have decided to maintain the existing format of the results section.
Comment: The design of the various clusters is quite heterogenous, at least according to Figure 3. Could you explain these differences?
Response: The heterogeneity in cluster design arises from geographical differences. Our study covered union councils in 10 high-risk districts from three different provinces (i.e. Karachi in Sindh, Khyber Pakhtunkhwa (KP), and Quetta block districts in Baluchistan). Despite this geographic diversity, the sociodemographic characteristics of enrolled children and their families were comparably uniform across all three areas, as detailed in Table 1a.
Comment: I did expect Table 5 and its content as the core of this study; not only a specific subheading should be therefore identified, but the content of this table should be more extensively reported and discussed.
Response: Thank you for your valuable comment. Indeed, the primary results for this study were presented in table 5 (now presented in Table 3 and Table 4. However, our discussion focuses on variables that demonstrated significance in the multivariate model, specifically those with a p-value < 0.1.
Comment: 5) Discussion could benefit from reporting of similar experience in other low income countries where vaccination rates were difficult to achieve because of the difficult acceptance by some population groups.
Response: Thank you for the feedback. We have added some relevant comparisons to the discussion section in addition to existing discussion.
Comments on the Quality of English Language
Comment: To be honest, the overall quality of the English text is more than appropriate and acceptable for publication. On the contrary, the text is affected by some typos, particularly in results section (e.g. rows 231 to 237 seem affected by some deletions and repetition, please double check).
Response: Thank you for the positive feedback. We addressed the typo.
Round 2
Reviewer 1 Report
Comments and Suggestions for Authors
Thank you for revising the paper. The improvements are substantial, and I am in agreement to accept the manuscript.
Reviewer 3 Report
Comments and Suggestions for Authors
Authors addressed all comments. The paper can now be accepted.
Reviewer 4 Report
Comments and Suggestions for Authors
Estimated authors,
I'm happy to share with you that the paper has been radically improved and, by my opinion, could be accepted for publication.
Some minor typos still remain (e.g. "A child is considered fully vaccinated s/he has..." did you mean "... IF s/he has..."?) but can be fixed during the pre-publishing stage.
Comments on the Quality of English LanguageSome minor typos still remain (e.g. "A child is considered fully vaccinated s/he has..." did you mean "... IF s/he has..."?) but can be fixed during the pre-publishing stage.